# The *foraging* gene coordinates brain and heart networks to modulate socially cued interval timing in *Drosophila*

Hongyu Miao☯, Wengjing Li☯, Yongwen Huang, Woo Jae Kim🆔*

HIT Center for Life Sciences, School of Life Science and Technology, Harbin Institute of Technology, Harbin, China

☯ These authors contributed equally to this work.
* wkim@hit.edu.cn

## Abstract

The *foraging* gene (*for*) regulates behavioral plasticity and decision-making, influencing adaptive behaviors such as foraging, learning, and memory. In *Drosophila melanogaster*, we explore its role in interval timing behaviors, particularly mating duration. Two allelic variants, rover (*for^R*) and sitter (*for^S*), exhibit distinct effects: *for^R* disrupts shorter mating duration (SMD) but not longer mating duration (LMD), while *for^S* impairs LMD but not SMD. Transheterozygotes (*for^R*/*for^S*) disrupt both behaviors, revealing complex allelic interactions. Using single-cell RNA sequencing and knockdown experiments, we identify *foraging* expression in *Pdfr*-positive neurons and *fru*-positive heart cells as critical for LMD. While the gene is expressed in memory-related brain regions, its impact on LMD is mediated through peptidergic signaling and calcium dynamics in the heart. Social context-dependent calcium fluctuations, observed via CaLexA signals, are disrupted by *foraging* or *Pdfr* knockdown, impairing LMD. These findings highlight the *foraging* gene's role in integrating social cues with physiological states. This study demonstrates the *foraging* gene's pleiotropic roles in regulating interval timing through neural and non-neural mechanisms, offering insights into the genetic and environmental interplay underlying adaptive behaviors.

## Author summary

This study reveals that the *foraging* gene, traditionally associated with behaviors such as foraging and learning, plays a critical role in interval timing (e.g., regulation of mating duration) in fruit flies (*Drosophila melanogaster*). Two natural alleles of the gene, rover (*forR*) and sitter (*forS*), specifically disrupt sexually experienced-induced shorter mating duration (SMD) and socially competitive-induced longer mating duration (LMD), respectively, while their transheterozygotes (*forR*/*forS*) impair both behaviors. Through single-cell RNA sequencing and functional

**Data availability statement:** All relevant data are in the manuscript and its supporting information files.

**Funding:** This research was supported a University of Ottawa Startup grant 602496 to WJK, Startup funds from HIT Center for Life Science to WJK, a University of Ottawa Interdisciplinary Research Group Funding Opportunity (IRGFO stream 1 and 2) grants 148101 and 148747 to WJK, a Natural Sciences and Engineering Research Council of Canada (NSERC) Discovery grant (reference: 211406) to WJK, a University of Ottawa Brain and Mind Research Institute/Center for Neural Dynamics Open call project grant 150950 to WJK, a Mitacs Globalink Research Internship Program grant 17268 to WJK. This research was also supported by the Brain Pool Program of the National Research Foundation in Korea grant ZYM5041911 to WJK, Burroughs Wellcome Fund Collaborative Research Travel Grants (reference: 1017486) to WJK and a NVIDIA Academic Hardware Grant Program to WJK. The funders had no role in study design, data collection and analysis, decision to publish, or preparation of the manuscript. HM received salary from the 'Startup funds from HIT Center for Life Science to WJK'.

**Competing interests:** The authors have declared that no competing interests exist.

experiments, the study identifies key roles for *foraging* in Pdfr-positive neurons of the brain's central complex and fru-positive cells in the heart. Although the gene is expressed in brain regions associated with memory, its effect on LMD depends on peptidergic signaling (via Pdfr) and calcium dynamics in the heart. Social context (e.g., group vs. isolated rearing) regulates calcium fluctuations in heart cells, a process disrupted by *foraging* or *Pdfr* knockdown, thereby impairing LMD. These findings uncover a novel mechanism by which the *foraging* gene integrates social cues with physiological states through neural and non-neural pathways, providing new insights into how genes regulate adaptive behaviors and their interactions with the environment via complex networks.

## Introduction

The *foraging* gene (*for*) has emerged as a pivotal factor in the regulation of behavioral plasticity and decision-making processes across various species, including the fruit fly *Drosophila melanogaster* [1,2] and honeybee [3]. This gene plays a crucial role in shaping an organism's foraging strategies by influencing traits such as olfactory learning and memory [4–10]. The study of the *foraging* gene has provided valuable insights into the complex interplay between genetics and behavior, highlighting its significance in understanding the adaptive behaviors of organisms in response to their environment [3,6].

Interval timing behaviors, the ability to measure and respond to the passage of time, are integral to various aspects of an organism's life, including foraging, mating, and social interactions [11,12]. These behaviors are thought to be mediated by neural circuits that are conserved across species [13]. In recent years, researchers have discovered a remarkable connection between interval timing behaviors and social behaviors, suggesting a shared neural and genetic basis [14].

The mating duration (MD) of male fruit flies, *Drosophila melanogaster*, serves as an excellent model for studying interval timing behaviors. In *Drosophila*, two notable interval timing behaviors related to mating duration have been identified: Longer-Mating-Duration (LMD), which is observed when males are in the presence of competitors and extends their mating duration [15–17] and Shorter-Mating-Duration (SMD), which is characterized by a reduction in mating time and is exhibited by sexually experienced males [18,19]. The MD of male fruit flies serves as an excellent model for studying interval timing, a process that can be modulated by internal states and environmental contexts. Previous studies by our group [15,16,19–22] and others [23–26] have established robust frameworks for investigating MD using advanced genetic tools, enabling the dissection of neural circuits and molecular mechanisms that govern interval timing.

The *foraging* gene emerged as a strong candidate for regulating LMD due to its well-documented role in behavioral plasticity and decision-making processes [6,27,28]. The *foraging* gene encodes a cGMP-dependent protein kinase (PKG), which has been implicated in modulating foraging behavior, aggression, and other

context-dependent behaviors in *Drosophila*. Its involvement in these processes suggests a potential role in integrating environmental cues and internal states to regulate interval timing, such as LMD. Furthermore, the molecular mechanisms underlying interval timing have been explored in other contexts, such as the work of the Crickmore et al. [29], which has demonstrated the critical role of CREB (cAMP response element-binding protein) in regulating behavioral timing and plasticity. CREB-dependent signaling pathways, along with other molecular players like PKG, provide a broader framework for understanding how interval timing is orchestrated at the neural and molecular levels [23–26,30,31]. By investigating *foraging* in the context of LMD, we aim to uncover how specific genetic and neural mechanisms fine-tune interval timing in response to social and environmental cues, contributing to a deeper understanding of the principles governing behavioral adaptation.

MD represents a critical investment of time for male reproductive success, highlighting its significance as a model for investigating interval timing. This parameter offers a valuable opportunity to explore the evolutionary and physiological roles of genes involved in this complex behavior [32]. From an evolutionary perspective, the ability to accurately regulate mating duration allows males to maximize their reproductive output by optimizing their investment of time and energy. This time investment strategy is crucial for competing with other males and ensuring successful fertilization of female eggs [19]. Physiologically, mating duration is influenced by a complex interplay of neural circuits, hormonal signals, and sensory inputs. Understanding the genes and mechanisms underlying this behavior can provide valuable insights into the neural substrates of interval timing and the regulation of reproductive behaviors [15,16,18,19,33–35].

The MD of male fruit flies is highly dependent on gene-environment interactions, making it a promising candidate to be regulated by the *foraging* gene function [15,16,19]. The *foraging* gene, known for its role in mediating behavioral plasticity and decision-making processes, is likely to influence mating duration by modulating the fly's response to environmental cues and social context. For instance, in the presence of competitors, the *foraging* gene may upregulate MD (LMD) to enhance reproductive success, while in situations of sexual satiation, it may downregulate MD (SMD) to conserve energy and resources. By examining the expression and function of the *foraging* gene in relation to MD, we can gain valuable insights into the intricate interplay between genetics, behavior, and environmental factors in *Drosophila*.

In this study, we demonstrate that genes traditionally associated with foraging behavior are also involved in interval timing, a cognitive process crucial for decision-making and behavior [12,36]. This discovery challenges the assumption that these genes are solely dedicated to optimizing foraging strategies and suggests a broader role for these genes in regulating time perception and behavior across various contexts [13,37]. Our findings indicate that the interplay between *foraging* genes and interval timing circuits may be crucial for animals to adapt their behavior to changing environmental conditions and maximize their overall fitness. This suggests that the evolution of interval timing and foraging behaviors might be more interconnected than previously thought [38,39].

Understanding the mechanisms by which *foraging* genes influence interval timing can provide valuable insights into the neural and genetic basis of time perception and decision-making. This knowledge has the potential to contribute to our understanding of various behaviors, including foraging, learning, and memory, and their adaptive significance in different ecological contexts.

## Results

### Two distinct *foraging* allele, rover and sitter affect distinct interval timing behaviors

The *foraging* gene in *Drosophila* gives rise to two distinct phenotypes known as rover (*for$^R$*) and sitter (*for$^S$*), which exhibit natural variations in foraging behavior. Rover larvae cover larger areas within food patches and greater movement between patches than sitters. Rover larvae and adults are more active than their sitter counterparts. This behavioral variation is primarily due to allelic differences in the *foraging* gene, which is located on the second chromosome and encodes a cGMP-dependent protein kinase (PKG). Natural rover (*for$^R$/for$^R$*) and sitter (*for$^s$/for$^s$*) strains exhibit allelic differences in *foraging* gene expression, with rovers showing elevated PKG activity compared to sitters due to the hypomorphic nature

of the $for^s$ allele [40]. The kinase encoded by *foraging* is a critical regulator of numerous downstream targets, leading to a range of pleiotropic effects associated with the *foraging* gene [1,41–43].

Moreover, the cultured giant neural characteristics of these phenotypes are distinctly different [42]. Neurons from sitter-allele homozygotes ($for^s/for^s$) display higher levels of spontaneous activity and exhibit exaggerated responses to stimulation, known as excessive evoked firing, which are not observed in neurons from rover-allele homozygotes ($for^R/for^R$). This heightened excitability in neurons of *sitter* allele is attributed to a reduction in voltage-dependent potassium ($K^+$) currents and the presence of excitable synapses. Additionally, the axon terminal projections of *sitter* strains are altered, further highlighting the neural basis for the behavioral differences between these two foraging strategies [42]. The *foraging* gene's pleiotropic roles in synaptic structure, vesicle dynamics, and neuronal-glial interactions underscore its importance in balancing energy allocation and synaptic plasticity. These findings provide a framework for understanding how genetic variation in *for* (PKG) shapes adaptive behaviors through NMJ remodeling [44,45].

The $for^R$ homozygote allele displayed deficiencies exclusively in SMD behavior (sup 1A), while the $for^S$ homozygote allele exhibited deficits only in LMD behavior (Fig 1B). Strikingly, the $for^R/for^S$ transheterozygote exhibited deficits in both LMD and SMD behaviors (Fig 1C), indicating that each allele specifically affects distinct interval timing behaviors. All adult flies that are heterozygous for the control demonstrate typical LMD behavior (S1A and S1B Fig). Therefore, each $for^R$ or $for^S$ allele displays a dominant phenotype when paired with another $for^R$ or $for^S$ allele (Fig 1C), but this dominance is lost when they are paired with a wild-type for allele (S1A and S1B Fig). In molecular terms, these findings indicate that the PKG activity and regulatory mechanisms associated with each *foraging* homozygous allele is essential for disrupting LMD or SMD behaviors. Therefore, we hypothesize that an extremely high level of PKG activity specifically disrupts SMD, while an extremely low level of PKG activity specifically disrupts LMD behavior.

Given the more pronounced defects associated with the sitter allele which shows lower PKG activity [41,42,46], we have chosen to concentrate our efforts on elucidating the genes and neural circuits that underlie the influence of the *foraging* gene on LMD behavior, with the aim of mapping the precise mechanisms governing this aspect of interval timing.

### Memory circuitry for LMD behavior is not linked to the function of *foraging*

To recreate the effects of the $for^S$ allele, we employed RNA interference (RNAi)-mediated gene knockdown. This technique lowers the expression of the PKG gene, consequently reducing PKG activity in cells expressing GAL4. The neuronal knockdown of *for* using two distinctive RNAi strains disrupted LMD behavior (Figs 1D and S1C), however glial knockdown had no effect (Fig 1E). Utilizing the cutting-edge fly SCope single-cell RNA-sequencing data platform [47], we detected a significant overlap in the expression of the *foraging* gene with two key markers: *elav*, indicative of neuroblast activity, and *nSyb*, a synaptic marker, within the neuronal population (Fig 1F). This co-expression pattern implies that the *foraging* gene likely plays a role in neuronal function. The *foraging* gene also exhibits a strong expression correlation with *repo*, a marker for glial cells, and is prominently expressed in specific regions of the central nervous system (CNS) surface glia (Fig 1G). Given that RNAi and GAL4 control cross flies exhibit normal LMD behavior (S1D and S1E Fig), we infer that the neuronal function of the *foraging* gene is indispensable for the generation of LMD behavior, whereas the glial function is not. Knockdown of *foraging* using the *nSyb-GAL4* driver, located on the third chromosome, also disrupted LMD behavior, confirming that the chromosomal location of the GAL4 transgene does not influence the observed effects of *foraging* knockdown (S1G Fig).

Given the *foraging* gene's established role in learning and memory in *Drosophila* [48,49], we investigated its function in key brain regions associated with memory. We have previously demonstrated that the neural circuits responsible for the formation and retention of longer mating duration (LMD) memories are located within the R2-R4m ring neurons of the ellipsoid body (EB) [15]. Knockdown of *for* in the mushroom body (MB) (Fig 2A), fan-shaped body (FB) (Fig 2B), EB (Fig 2C), and pars intercerebrails (PI) (Fig 2D) did not affect LMD behavior, indicating that *for* expression in these memory-related brain regions is dispensable for LMD generation. This aligns with previous findings that MB ablation via

PLOS Genetics

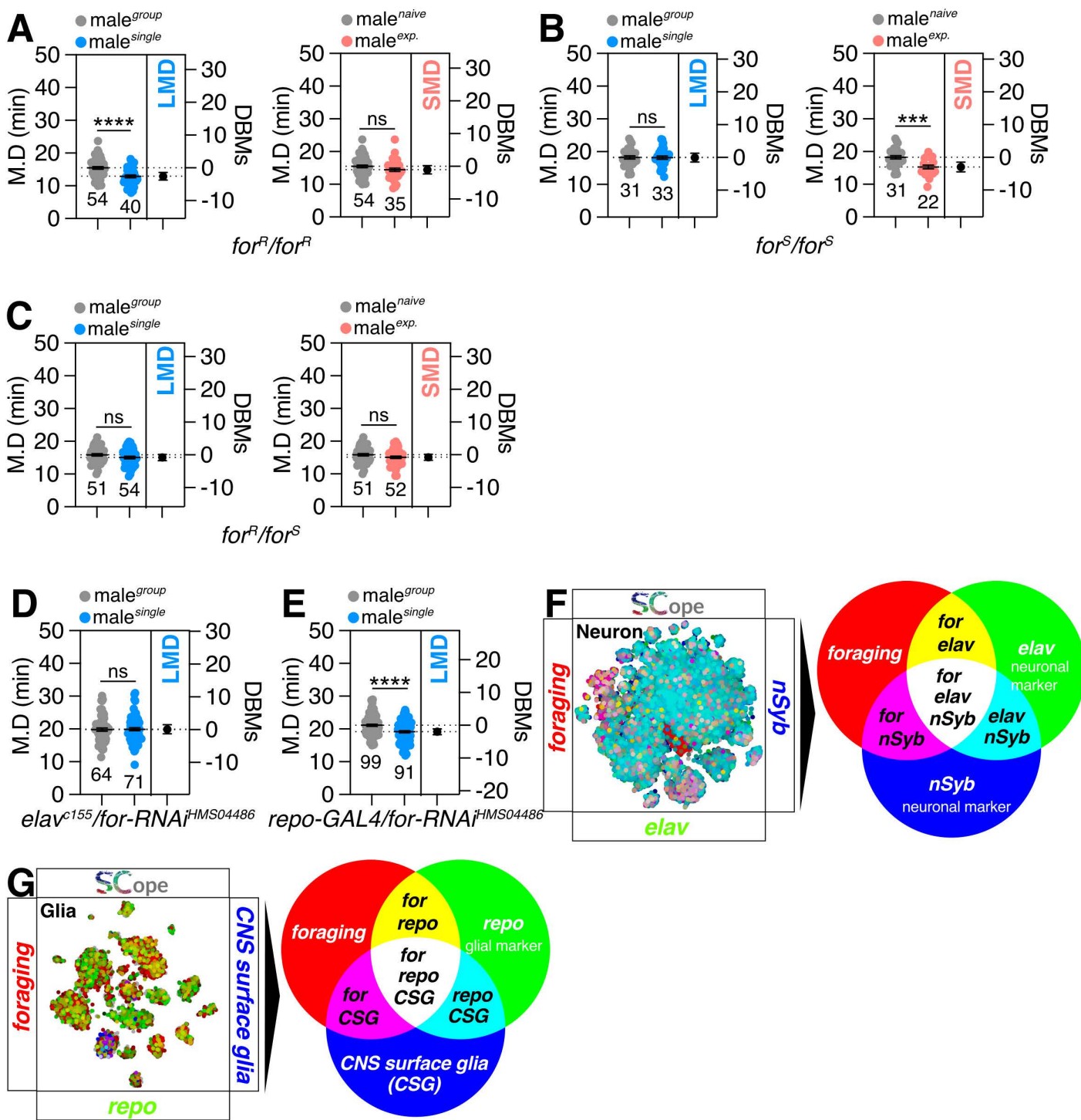

**Fig 1. Interval behavior is regulated by two distinct *foraging* alleles. (A-C)** LMD and SMD assays for *for^R* homozygous variants, *for^S* homozygous variants and transheterozygote *for^R/for^S*. Light grey dots represent group males and blue dots represent single reared ones. Dot plots represent the MD (Mating Duration) of each male fly. DBMs represent difference between means. The mean value and standard error are labeled within the dot plot (black lines). Asterisks represent significant differences, as revealed by the Student's t test (* p < 0.05, ** p < 0.01, *** p < 0.001). The same notations for statistical significance are used in other figures. **(D)** LMD assay of flies expressing *elav^c155* (neuron) driver together with *for-RNAi*. **(E)** LMD assay of flies expressing *repo-GAL4* (glia) driver together with *for-RNAi*. **(F)** Single-cell RNA sequencing (SCOPE scRNA-seq) datasets reveal cell clusters colored by

expression of *foraging* (red), *nSyb*/*elav* (blue/green) in neurons. **(G)** SCOPE scRNA-seq datasets reveal cell clusters colored by expression of *foraging* (red), CNS surface glia/repo (blue/green) in glia cells.

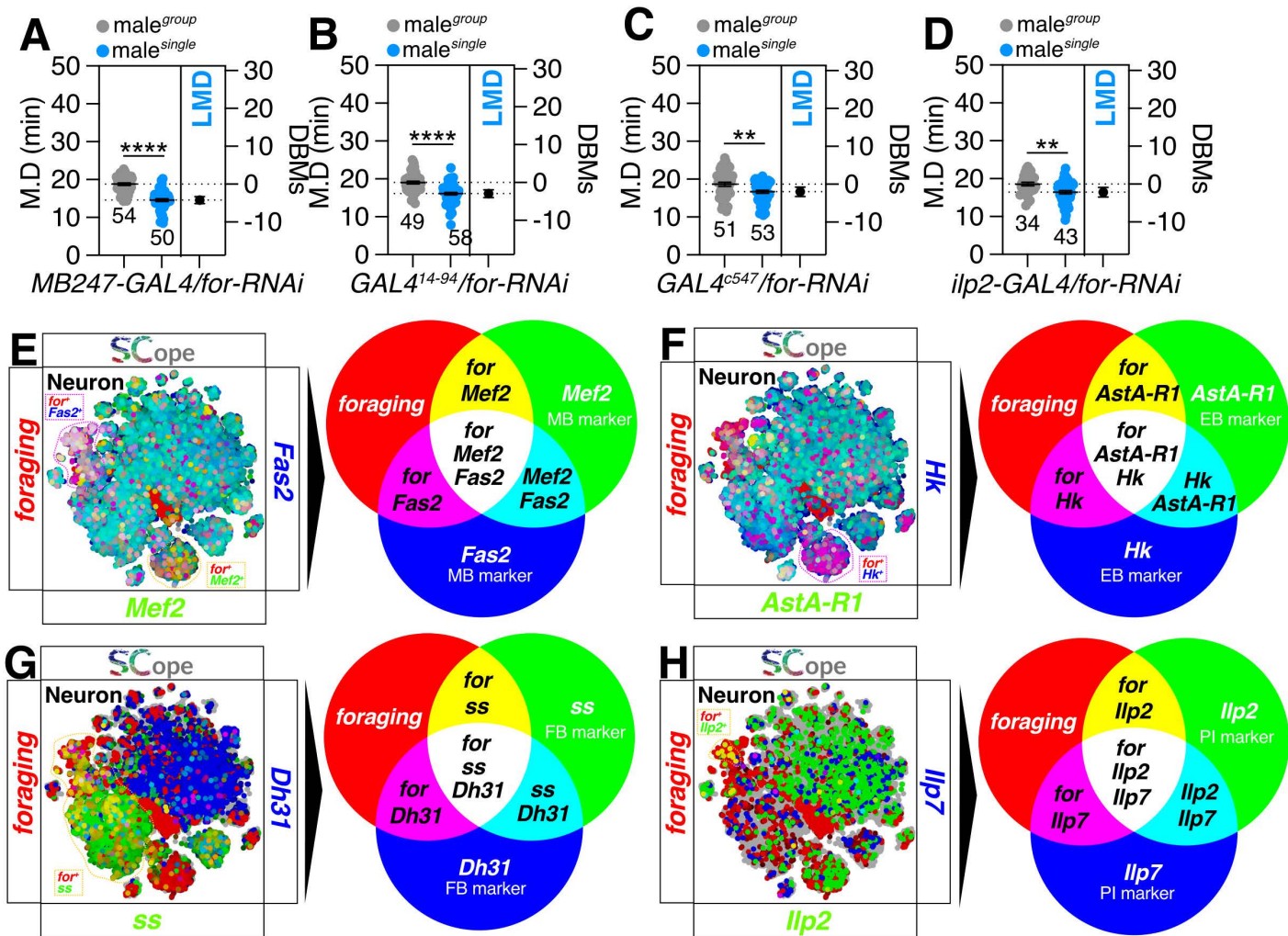

**Fig 2. The memory circuitry responsible for LMD behavior does not correlate with *foraging* activities. (A)** LMD assay of flies expressing *MB247-GAL4* (MB) driver together with *for-RNAi*. **(B)** LMD assay of flies expressing *GAL4*[14-94](FB) driver together with *for-RNAi*. **(C)** LMD assay of flies expressing *c547-GAL4* (EB) driver together with *for-RNAi*. **(D)** LMD assay of flies expressing *Ilp2-GAL4* (PI) driver together with *for-RNAi*. **(E)** SCOPE scRNA-seq datasets reveal cell clusters colored by expression of *foraging* (red), *Fas2*/*Mef2* (blue/green) in glia cells. **(F)** SCOPE scRNA-seq datasets reveal cell clusters colored by expression of *foraging* (red), *Hk*/*AstA-R1* (blue/green) in neurons. **(G)** SCOPE scRNA-seq datasets reveal cell clusters colored by expression of *foraging* (red), *Dh31*/*ss* (blue/green) in glia cells. **(H)** SCOPE scRNA-seq datasets reveal cell clusters colored by expression of *foraging* (red), *Ilp7*/*Ilp2* (blue/green) in glia cells.

hydroxyurea feeding did not alter the locomotor activity of rover and sitter larval morphs [50]. Our findings reveal those molecular markers for the MB (Fas2- and Mef2-positive) [51,52], EB (Hk- and AstA-R1-positive) [53,54], FB (Dh31- and ss-positive) [55,56], and PI (Ilp2-positive but Ilp7-negative) [57] specifically overlap with *foraging* gene expression in certain neuronal populations (Fig 2E–H). This suggests that *foraging* gene expression within these memory-related brain

regions is not essential for the formation of memories underlying LMD behavior, contrary to the previous assumption that such memories were primarily localized to the EB rather than the MB or FB [15]. Therefore, the *foraging* gene may not be directly involved in the generation of LMD-related memories, or there may be other, as yet unidentified, brain regions critical for this function.

## A distinct population of peptidergic neurons that express the *for* gene are responsible for regulating LMD

LMD behavior is known to depend on the peptidergic circuitry of PDF/NPF, but not sNPF [16]. However, our data show that *foraging* gene expression in these peptidergic neurons does not affect LMD behavior, nor is *foraging* gene expression detected in these circuits (Fig 3A–F). This indicates that *foraging* gene expression in the peptidergic neurons is also not required for LMD behavior associated with *foraging* gene function. Despite the absence of a clear co-expression pattern between *foraging* gene and *Pdf* or *NPF* in neuronal populations, we observed a significant overlap between the expression of *sNPF* and *foraging* gene (Fig 3F).

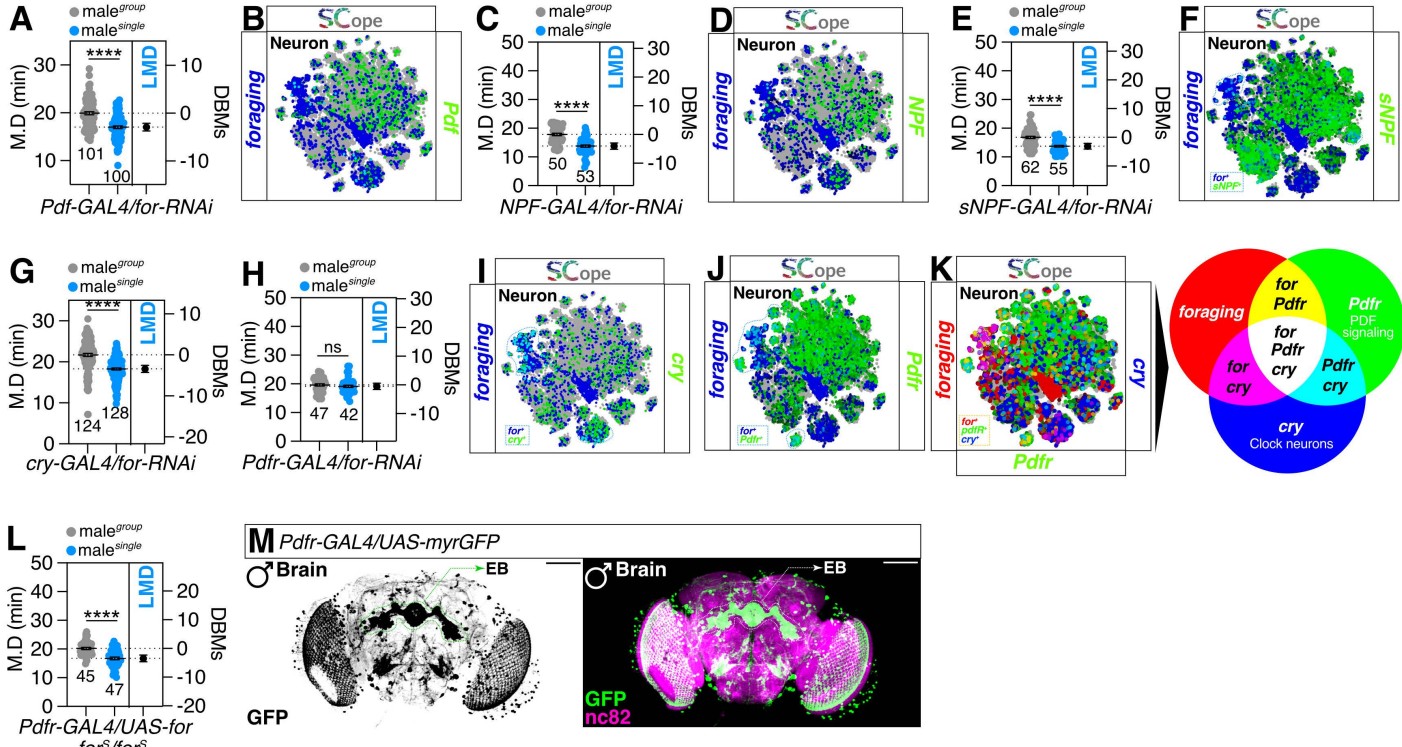

**Fig 3. Pdfr neurons that express the *for* gene are responsible for regulating LMD. (A)** LMD assay of flies expressing *Pdf-GAL4* driver together with *for-RNAi*. **(B)** SCOPE scRNA-seq datasets reveal cell clusters colored by expression of *foraging* (blue), *Pdf* (green) in neurons. **(C)** LMD assay of flies expressing *NPF-GAL4* driver together with *for-RNAi*. **(D)** SCOPE scRNA-seq datasets reveal cell clusters colored by expression of *foraging* (blue), *NPF* (green) in neurons. **(E)** LMD assay of flies expressing *sNPF-GAL4* driver together with *for-RNAi*. **(F)** SCOPE scRNA-seq datasets reveal cell clusters colored by expression of *foraging* (blue), *sNPF* (green) in neurons. **(G)** LMD assay of flies expressing *cry-GAL4* driver together with *for-RNAi*. **(H)** LMD assay of flies expressing *Pdfr²ᴬ-GAL4* driver together with *for-RNAi*. **(I)** SCOPE scRNA-seq datasets reveal cell clusters colored by expression of *foraging* (blue), *cry* (green) in neurons. **(J)** SCOPE scRNA-seq datasets reveal cell clusters colored by expression of *foraging* (blue), *Pdfr* (green) in neurons. **(K)** SCOPE scRNA-seq datasets reveal cell clusters colored by expression of *foraging* (red), *cry/Pdfr* (blue/green) in neurons. **(L)** LMD assay for *Pdfr²ᴬ-GAL4* drives *for* overexpression under *forˢ* homozygote. **(M)** Male flies brain expressing the *Pdfr²ᴬ-GAL4* together with *UAS-myrGFP* were immunostained with anti-GFP (green) and nc82 (magenta) antibodies. Scale bars represent 100 μm.

Given that Pdf signaling is known to operate through cry- and Pdfr-positive clock neuronal circuits [16], we investigated the role of *foraging* gene within these circuits. We discovered that knocking down *foraging* gene expression in Pdfr-expressing cells, but not cry-expressing cells impairs LMD behavior (Fig 3G–H), and that *foraging* gene is expressed in a specific neuronal population where *cry* and *Pdfr* are expressed (Fig 3I–K). Furthermore, genetically restoring wild-type *foraging* gene expression in Pdfr-positive cells of the *for^s* variant background rescues LMD behavior (Figs 3L and S1F), indicating that Pdfr-positive cells are a crucial population for the generation of LMD behavior in the context of *foraging* gene function.

## Expression of *foraging* in Pdfr-expressing specific cell population is essential for the induction of LMD behavior

Upon examining the expression patterns of Pdfr-positive cells in the brain, we observed robust expression within the EB (Fig 3M). Our previous research also suggested that the EB is a central region involved in forming memories related to LMD [15]. To identify the precise region within the EB where the *foraging* gene is active in promoting LMD behavior, we employed a mini-scale screening approach using established enhancer trap GAL4 lines [58]. These lines have been effectively utilized to delineate the anatomical structure of the adult central complex. We identified cells labeled by *30y-* and *c61-GAL4* drivers as potential candidates (Fig 4A–J). The *30y-GAL4* driver is known for its expression in the MB, EB and FB, subesophageal ganglion (SOG), antennal & optic lobes (AL & OL), protocerebrum, and median bundle. Meanwhile, *c61-GAL4* has been associated with expression in Fa3 neurons of the central complex (CC) and extends projections through the protocerebral bridge (PB) to the FB [58–60]. Given that the *c61-GAL4* driver, which is specific to a restricted set of neurons, disrupted LMD behavior when *for* gene expression was knocked down (Fig 4J), while the broader EB driver *c547-GAL4* did not (Fig 2C), we hypothesize that a very specific subset of c61-positive neurons that are c547-negative are solely responsible for the neuronal function of *foraging* that leads to LMD.

Screening for overexpression of the *foraging* gene has provided evidence that the expression level of *foraging* in cells targeted by the *30y-GAL4* and *c61-GAL4* drivers is critical for the generation of LMD (S2A–K Fig). This finding aligns with previous research indicating that feeding-related traits are influenced by the dosage of the *foraging* gene [40]. By expressing wild-type *for* gene in cells labeled by the *30y-GAL4* driver, we were able to restore normal LMD behavior (S2L Fig). Moreover, utilizing a recently identified EB-split GAL4 driver line (SS00096) [61], we demonstrated that knocking down *for* expression in specific EB neurons significantly impacts LMD behavior (Fig 4K–L). Aligning with genetic control data (S2M Fig), our results further suggest that the interval timing-related traits are also modulated by the dosage of the *foraging* gene within specific cell populations of the EB.

## *foraging* expression in the heart modulates calcium dynamics via *Pdfr* signaling to regulate LMD behavior

Although the specific population of neurons in the EB relies on the *foraging* gene for the induction of LMD, there may also be a non-neuronal contribution from the *foraging gene* to elicit LMD behavior. This is supported by our recent findings that corazonin receptor (CrzR), a gene expressed in glial cells, is a crucial component for LMD [62]. Recent studies have indicated that the *foraging* gene is expressed across various tissues and is not confined to neuronal populations, extending its reach to glial subtypes and peripheral tissues such as the gastric and reproductive systems [63]. Additionally, the expression of the *foraging* gene in multiple tissues and organs is known to be influenced by starvation [10]. Utilizing the fly SCope 10x-Cross tissue platform, we discovered that the *foraging* gene is expressed at high levels in the majority of tissues (S3A–N Fig). Notably, the *foraging* gene shows particularly strong co-expression with *Pdfr* in the fat body (S3A Fig), muscle (S3I Fig), and oenocytes (S3J Fig). Using the 10x and SMART-seq2 platforms, we identified robust co-expression of *foraging* and *Pdfr* in the heart (Fig 5A).

Our findings implicate PDF-PDFR mediated peptidergic signaling as a mechanism through which this specific organ system influences a diverse array of behaviors and physiological processes. Considering the critical role of these organs

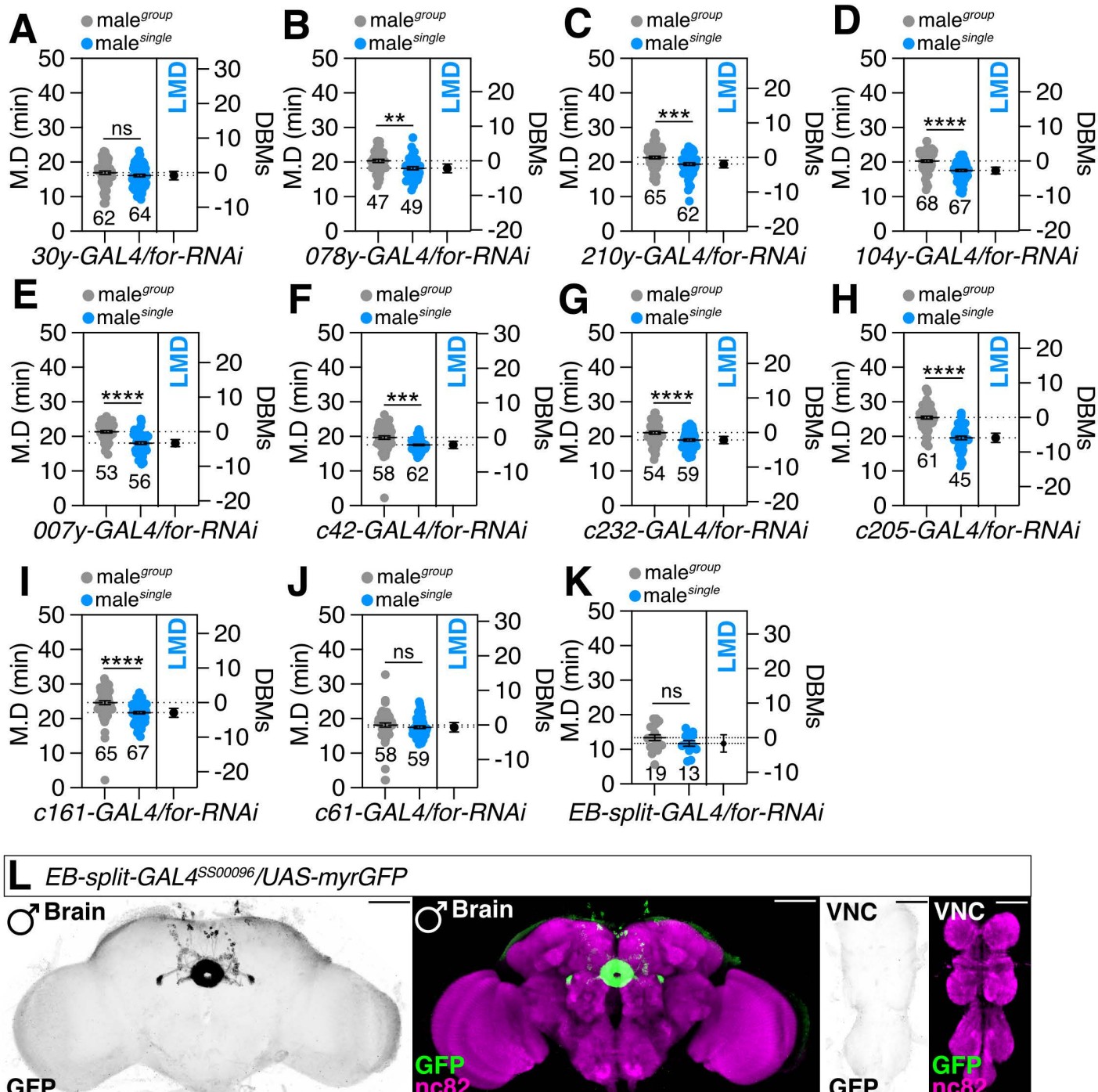

**Fig 4. Candidate Pdfr-expressing cells in the ellipsoid body drivers for LMD screening. (A-J)** LMD assays for male knockdown of *foraging* driven by subsets of neuronal cell lines of the EB, *30y-GAL4*(A), *078y-GAL4* **(B)**, *210y-GAL4* **(C)**, *104y-GAL4* **(D)**, *007y-GAL4* **(E)**, *c42-GAL4* **(F)**, *c232-GAL4* **(G)**, *c205-GAL4* **(H)**, *c161-GAL4* **(I)**, and *c61-GAL4* **(J)**. **(K)** LMD assay for male knockdown of foraging driven by *EB-split-GAL4*, SS00096. **(L)** Male flies expressing *EB-split-GAL4*, SS00096, with *UAS-myrGFP*. Source image lsm files (~500MB) were downloaded from FlyLight platform constructed by Janelia Farm Research Center (JFRC) then reconstructed using ImageJ https://flylight-raw.janelia.org/cgi-bin/view_raw_imagery.cgi?line=SS00096 [96, 97]. Scale bars represent 100 μm.

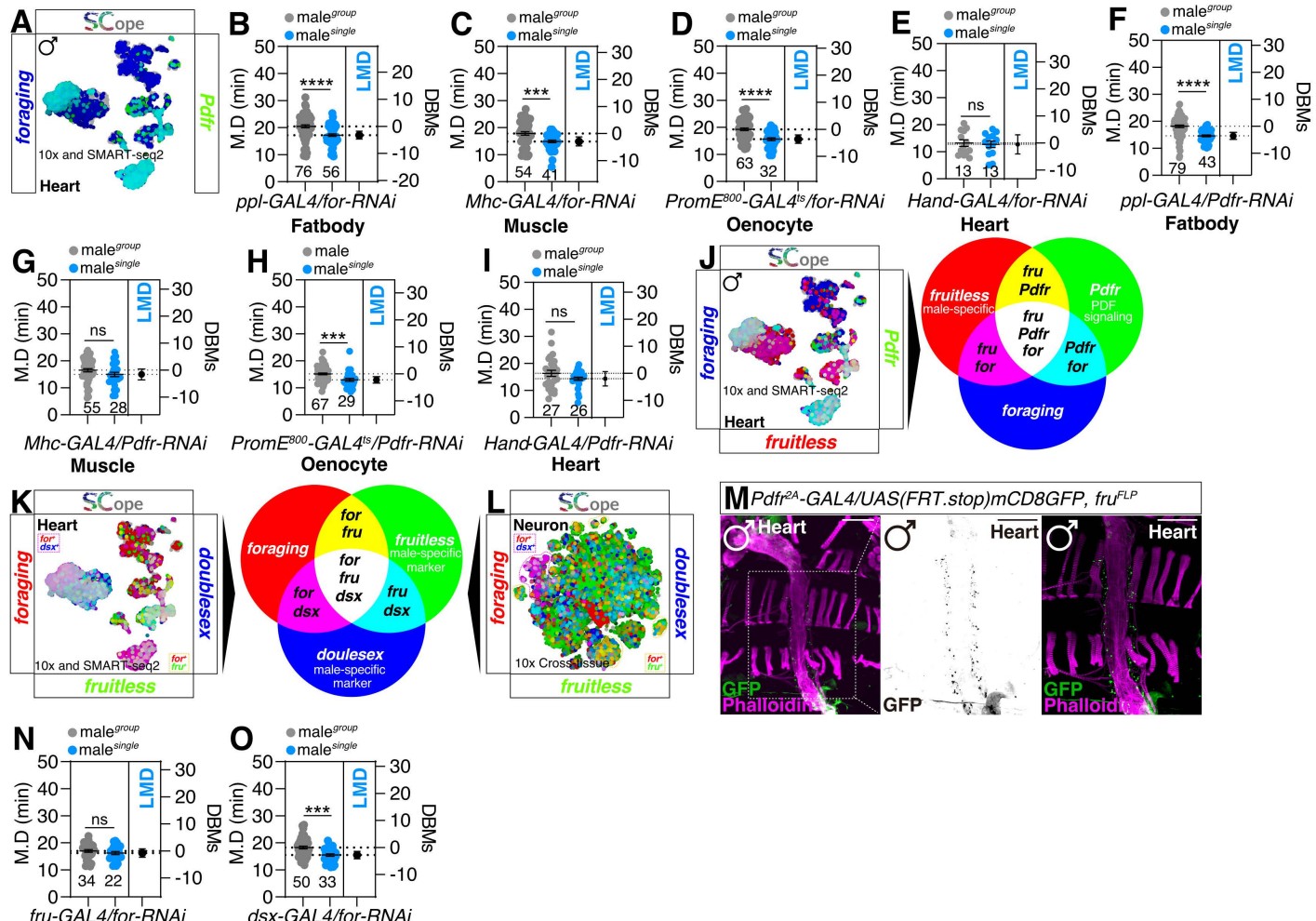

**Fig 5. Sexually dimorphic expression of the *foraging* gene in the heart. (A)** SCOPE scRNA-seq datasets reveal cell clusters colored by expression of *foraging* (blue), *Pdfr* (green) in heart. **(B-E)** LMD assays for tissue-specific knockdown of *foraging* via *for-RNAi* using (B) *ppl-GAL4*, **(C)** *Mhc-GAL4*, **(D)** *PromE800-GAL4ts* and **(E)** *Hand-GAL4*. **(F-I)** LMD assays for tissue-specific knockdown of *Pdfr* via *Pdfr-RNAi* using (F) *ppl-GAL4*, **(G)** *Mhc-GAL4*, **(H)** *PromE800-GAL4ts* and **(I)** *Hand-GAL4*. **(J)** SCOPE scRNA-seq datasets reveal cell clusters colored by expression of *foraging* (blue), *fruitless/Pdfr* (red/green) in heart. **(K-L)** SCOPE scRNA-seq datasets reveal cell clusters colored by expression of *foraging* (red), *doublesex/fruitless* (blue/green) in heart (K) and neurons **(L)**. **(M)** Male flies heart expressing the *UAS(FRT.stop)mCD8GFP; fruFLP* together with *Pdfr2A-GAL4* were immunostained with anti-GFP (green) antibody and phalloidin (magenta). **(N)** LMD assay for male knockdown of *foraging* driven by *fru-GAL4*. Scale bars represent 100 μm. **(O)** LMD assay for male knockdown of *foraging* driven by *dsx-GAL4*.

in sugar and lipid metabolism, essential for energy storage and homeostasis, it is plausible that *foraging* gene expression within these tissues may significantly regulate energy-related behaviors and physiological responses [64–68]. Notably, the knockdown of *foraging* gene expression in the fat body, muscle, and oenocytes did not perturb LMD behavior (Fig 5B–D). However, targeted knockdown of *foraging* in the heart using the *Hand-GAL4* driver resulted in disrupted LMD behavior (Fig 5E). Similarly, knockdown of *Pdfr* in both muscle and heart tissues also led to impaired LMD (Fig 5F–I), indicating that *Pdfr* expression in the heart and muscle is essential for activating the signaling pathways required to generate LMD behavior. These results indicate that *foraging* gene expression in the heart (S3O Fig) is essential for the maintenance of interval timing behaviors.

## Interval timing generation depends on *foraging* function in *fru*-positive heart cells

LMD represents a male-specific interval timing behavior that relies on sexually dimorphic *NPF* neurons situated within the brain [15,16]. In *Drosophila*, sex-biased gene expression in the brain can indeed represent sexual dimorphic features [69]. Sexual dimorphism refers to the phenotypic differences between males and females of the same species. These differences can be influenced by genetic factors, including differential expression of genes between the sexes [70,71].

In *Drosophila*, sexual dimorphism refers to the distinct differences in phenotype between males and females. These differences are largely determined by the expression of the *fruitless (fru)* and *doublesex (dsx)* genes, which play critical roles in the development of male and female characteristics. These genes regulate the development of male and female-specific traits, contributing to the distinct phenotypes observed between males and females of the same species [72–76].

Investigation of the co-expression patterns of the *foraging* gene with *fru* and *dsx* revealed distinct heart cell and neuronal populations that express foraging in conjunction with either *fru* or *dsx* (Fig 5J–L). The *fru* gene is primarily involved in the regulation of sexual behaviors through the modulation of neural pathways, whereas *dsx* exhibits broader effects, impacting both neuronal circuits and the development of sexual characteristics in various cell types, extending beyond neurons [73]. Indeed, we observed *Pdfr* expression in both *fru*-positive neurons and heart cells (Figs 5M and S3P). Notably, *Pdfr*-positive brain neurons did not overlap with the EB, consistent with previous reports that EB cells are not fru-positive [77,78]. Additionally, we found that the *30y-GAL4* driver labels *fru*-positive heart cells, similar to the *Pdfr²ᴬ-GAL4* driver (S3Q Fig). Specifically, the inhibition of the *for* gene in *fru*-expressing cells resulted in the disruption of LMD behavior (Fig 5N), whereas the knockdown of the *for* gene in *dsx*-expressing cells did not affect LMD behavior (Fig 5O).

Therefore, we conclude that the knockdown and genetic rescue effects observed with the *Pdfr²ᴬ-GAL4* driver (Fig 3J and 3L) and the *30y-GAL4* driver (Figs 4A, S2A, and S2L) are attributable to their expression in the heart. In summary, our findings demonstrate that *fru*-positive heart cells expressing *foraging* and *Pdfr* play a critical role in mediating LMD behavior. These findings support the hypothesis that foraging exhibits sexually dimorphic effects primarily in male-specific neuronal populations.

## Heart calcium dynamics are regulated by *foraging* and *Pdfr* signaling in a social context-dependent manner

Given the compelling evidence indicating that foraging function is exclusively dependent on a specific neuronal population and not on other tissues for the manifestation of LMD behavior, we sought to explore the calcium response properties of *for*^MI01791-TG4.1 neurons in flies experiencing diverse social conditions. Utilizing CaLexA, a transcription-based calcium reporter system [79], we observed elevated levels of CaLexA signals originating from *for*-positive neurons within the central brain region of group-reared male flies compared to those reared in isolation (S4A–C Fig). In contrast, the levels of CaLexA signals within the fat body remained unchanged between socially isolated and group-reared flies (S4D–E Fig). This differential response suggests that calcium signaling within a specific neuronal population proximal to the central brain region plays a pivotal role in modulating interval timing behavior.

Calcium dynamics play a pivotal role in regulating heart function across species, from humans to fruit fly [80–82]. In humans, calcium ions ($Ca^{2+}$) are central to cardiac muscle contraction and relaxation, acting as key mediators of excitation-contraction coupling. In response to electrical signals, $Ca^{2+}$ influx through voltage-gated channels triggers the release of stored calcium from the sarcoplasmic reticulum, leading to muscle contraction. Dysregulation of calcium handling is implicated in various cardiac pathologies, such as arrhythmias and heart failure [82]. We observed that fly heart pericardial cells (PCs) exhibit robust CaLexA signals under group-housed conditions but not in socially isolated conditions (Fig 6A–C). These changes in calcium dynamics were abolished when either *foraging* (Fig 6D–E) or *Pdfr* (Fig 6F–G) was knocked down in the heart, indicating that both genes are essential for mediating social context-dependent calcium signaling in PCs.

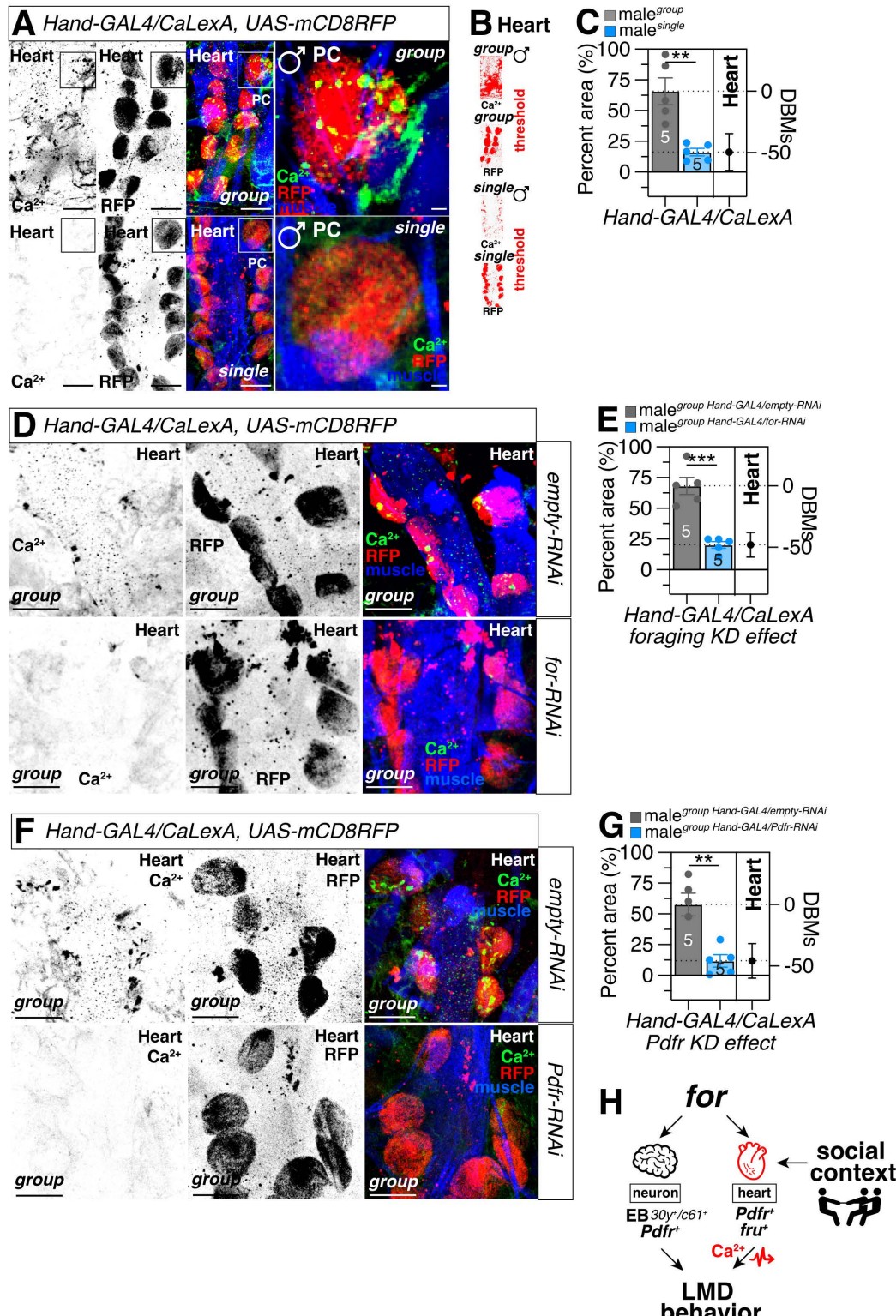

**Fig 6. The dynamics of heart calcium are modulated by *foraging* activities and *Pdfr* signaling. (A)** CalexA assay for *Hand-GAL4* together with l*exAop-mCD8GFP; UAS-CaLexA, lexAop-CD2-GFP* of group (up) and single (bottom) male flies. The box represents the fluorescence signal of an individual pericardial cell (PC). Scale bars represent 50 μm in heart images and represent 5 μm in individual PC images. See the "Methods" for a detailed

description of the fluorescence intensity analysis used in this study. **(B)** The small panels are presented as a red scale to show the GFP/RFP fluorescence marked by the threshold function of ImageJ. **(C)** Quantification of percent area for calcium signals displayed in (A) between group and single male flies. **(D)** CalexA assay for *Hand-GAL4* together with *UAS-mCD8RFP, lexAop-mCD8GFP;* l*exAop-mCD8GFP; UAS-CaLexA, lexAop-CD2-GFP* of genetic control (*empty-RNAi*, up) and group male flies with knocking down *foraging* (*for-RNAi*, bottom). RFP (*UAS-mCD8RFP*) labels cell membranes independently of CaLexA-GFP. Strain generated by integrating BDSC #32229 and #66542. Scale bars represent 50 μm in heart images. **(E)** Quantification of percent area for calcium signals displayed in (D) between group male flies expressing *Hand-GAL4* with *empty-RNAi* and *for-RNAi*. **(F)** CalexA assay for *Hand-GAL4* together with l*exAop-mCD8GFP; UAS-CaLexA, lexAop-CD2-GFP* of genetic control (*empty-RNAi*, up) and group male flies with knocking down *Pdfr* (*Pdfr-RNAi*, bottom). Scale bars represent 50 μm in heart images. **(G)** Quantification of percent area for calcium signals displayed in (F) between group male flies expressing *Hand-GAL4* with *empty-RNAi* and *Pdfr-RNAi*. **(H)** Model of how *foraging* regulate LMD behavior. The images are from https://www.svgrepo.com/.

Previous studies have delineated the multifaceted influence of the *foraging* gene, extending beyond locomotor behavior to include alcohol sensitivity [7], sucrose responses [8], and social network formation [28]. These behaviors are all profoundly influenced by the serotonergic system, which plays a pivotal role in modulating a wide array of physiological and behavioral processes [83–87]. Despite this, our experiments indicate that the *foraging* gene's expression in the serotonergic system is not a prerequisite for LMD behavior. By manipulating *foraging* gene levels within the serotonergic system using *5-HT1B*-GAL4 and *Trhn-GAL4* drivers, we observed no disruption in LMD (S5A–D Fig). Furthermore, we did not detect a significant co-expression between the *foraging* gene and either *5-HT1B* or *Trhn* (S5E–5F Fig), suggesting that the serotonergic system is not essential for the *foraging* gene's function in generating LMD behavior.

Collectively, these findings indicate that *foraging* expression collaborates with PDF-PDFR signaling in specific EB brain regions and *fru*-positive heart cells to regulate calcium dynamics in a social context-dependent manner. This modulation of calcium dynamics ultimately influences LMD, an interval timing behavior (Fig 6H).

## Discussion

This study explores the role of the *foraging* gene in regulating interval timing behaviors, particularly mating duration, in *Drosophila melanogaster*. The *foraging* gene, known for its influence on behavioral plasticity and decision-making, exhibits distinct effects in two allelic variants—rover (*for^R*) and sitter (*for^S*). These variants display specific deficiencies in LMD and SMD behaviors, highlighting the gene's role in interval timing (Fig 1). While the gene is expressed in key brain regions associated with memory and learning, its impact on LMD was not observed in these areas (Fig 2). Instead, *foraging* expression in *Pdfr*-positive neurons and *fru*-positive heart cells was found to be critical for LMD, suggesting its involvement in both neural and non-neural circuits (Figs 3–5). Additionally, the study revealed that *foraging* modulates calcium dynamics in a social context-dependent manner, further linking its function to interval timing behaviors (Fig 6). These findings underscore the complex interplay between genetic, neural, and environmental factors in regulating interval timing, with the *foraging* gene playing a central role in orchestrating these behaviors.

The *foraging* gene exhibits distinct influences on interval timing behaviors (LMD and SMD) in *Drosophila*, with the rover variant specifically affecting SMD and the sitter variant affecting LMD. The rescue of LMD by reintroducing wild-type *foraging* gene expression into the appropriate neurons highlights the critical role of the *foraging* gene in mediating interval timing behaviors [8,40,46,63]. Given the utilization of distinct circadian clock genes, neuropeptides, and neural circuits by LMD and SMD [18,34], it would be intriguing to explore the mechanisms by which the single gene *foraging* can differentially modulate these two distinct interval timing behaviors.

The *foraging* gene plays a critical role in regulating interval timing behaviors, with its allelic variants, *rover* and *sitter*, exhibiting distinct effects on LMD and SMD. These differences are primarily driven by their opposing impacts on cGMP-dependent protein kinase (PKG) activity. The *for^R* allele, associated with higher PKG activity, disrupts SMD while maintaining normal LMD (Fig 1A), suggesting that elevated PKG levels may hyperactivate or desensitize neural circuits

specific to SMD processes. Conversely, the *for^S* allele, characterized by lower PKG activity, impairs LMD but not SMD (Fig 1B), indicating that reduced PKG activity fails to meet the neuromodulatory thresholds required for LMD coordination. The *for^R/for^S* transheterozygotes, which disrupt both LMD and SMD (Fig 1C), reveal a complex interaction between these alleles, likely due to conflicting PKG activity levels or metabolic and neuronal polymorphisms that destabilize shared pathways. This phenomenon underscores the *foraging* gene's pleiotropic roles, where allelic balance fine-tunes PKG activity to maintain behavioral robustness, while extreme or mismatched levels disrupt circuit-specific thresholds critical for distinct memory processes [6,10]. While our findings support PKG-mediated mechanisms, we acknowledge the alternative possibility that background mutations in the *for^R* or *for^s* stocks might contribute to the observed transheterozygotes phenotype. Such mutations could behave recessively in heterozygotes but show dominant interactions in the *for^R/for^S* combination. Future studies using CRISPR-generated alleles could help resolve these competing explanations.

The *foraging* gene's influence on interval timing behaviors extends beyond neural circuits to include metabolic and synaptic regulation. The intact behaviors observed in *for^R/+* or *for^S/+* heterozygotes suggest that intermediate PKG activity levels balance circuit dynamics, allowing for normal LMD and SMD. However, the dual deficits in *for^R/for^S* transheterozygotes highlight the importance of allelic balance, as conflicting PKG levels may lead to systemic disruptions in both metabolic and neural pathways. This aligns with previous studies showing that *foraging* mediates adult plasticity and gene-environment interactions, particularly under stress conditions, and regulates synaptic terminal morphology and neuronal excitability [27,42]. The gene's role in integrating genetic and environmental cues further emphasizes its central role in adaptive behaviors. Collectively, these findings illustrate the complex interplay between PKG activity, neural circuits, and metabolic regulation in shaping interval timing behaviors, highlighting the *foraging* gene as a key modulator of behavioral plasticity in *Drosophila* [3,6,27].

Our findings reveal a previously unrecognized non-neuronal mechanisms by which foraging modulates interval timing behaviors through its function in *fru*-positive heart cells. We demonstrate that *foraging* and *Pdfr* co-expression in these cells regulates social context-dependent calcium dynamics, as evidenced by robust CaLexA signals in group-housed flies, which are absent in socially isolated individuals (Figs 5A and 6A–C). Knockdown of either *foraging* or *Pdfr* in the heart disrupts these calcium fluctuations and impairs LMD behavior (Fig 5E–G), indicating that PDF-PDFR signaling acts in concert with *foraging* to modulate cardiac calcium activity. This interaction likely bridges social environmental cues with internal metabolic states, as *foraging* is known to interact with insulin signaling under stress, and PDF-PDFR pathways integrate circadian and peptidergic inputs. The sexually dimorphic nature of LMD, coupled with *fru*'s role in male-specific traits, suggests that *foraging*'s function in *fru*-positive heart cells represents a critical node for male reproductive strategies, linking cardiac physiology to interval timing behaviors. These results expand the scope of *foraging*'s pleiotropy, highlighting its role in coordinating organ-level calcium dynamics with neural circuits to adaptively regulate time-sensitive behaviors in response to social contexts.

In summary, the critical role of the *foraging* gene in mediating complex interval timing behaviors in *Drosophila* is evident. The gene's expression in specific neurons and heart cells is crucial for its function. The intricate interplay between genetics, environment, and behavior revealed by these findings has broader implications for understanding the regulation of complex behaviors across species.

## Methods

### Ethics statement

All animal experiments reported in this manuscript were conducted in compliance with the ARRIVE guidelines and adhered to the U.K. Animals (Scientific Procedures) Act, 1986 and associated guidelines, EU Directive 2010/63/EU for animal experiments, or the National Research Council's Guide for the Care and Use of Laboratory Animals.

## Fly stocks and husbandry

*Drosophila melanogaster* were raised on cornmeal-yeast medium at similar densities to yield adults with similar body sizes. Flies were kept in 12 h light: 12 h dark cycles (LD) at 25°C (ZT 0 is the beginning of the light phase, ZT12 beginning of the dark phase) except for some experimental manipulation (experiments with the flies carrying *tub-GAL80ts*). Wild-type flies were *Canton-S* (*CS*). We have previously demonstrated that our *CS* flies exhibit normal LMD and SMD behaviors [15,19]. To reduce the variation from genetic background, all flies were backcrossed for at least 3 generations to *CS* strain. Following lines used in this study, *Canton-S* (#64349), *Df(1) Exel6234*(#7708), *forS* (#76120), *elavc155* (#25750), *repo-GAL4*(#7415), *nSyb-GAL4*(#68222), *MB247-GAL4*(#50742), *c547-GAL4*(#30819), *Ilp2-GAL4*(#37516), *Pdf-GAL4*(#6899), *NPF-GAL4*(#25682), *sNPF-T2A-GAL4*(#84706), *empty-RNAi*(#36304), *cry-GAL4*(#24514), *Pdfr-GAL4*(#33070), *30y-GAL4*(#30818), *078y-GAL4*(#30821), *104y-GAL4*(#81014), *007y-GAL4*(#30812), *c42-GAL4*(#30835), *c232-GAL4*(#30828), *c205-GAL4*(#30826), *c161-GAL4*(#27893), *c61-GAL4*(#30845), *EB-split-GAL4SS00096*(#86861), *5-HT1B-GAL4*(#86276), *Trhn-GAL4*(#84694), *P{PZ}for[06860]*(#12326), *ppl-GAL4*(#58768), *Mhc-GAL4*(#55133), *PromE-GAL4*(#65404), *Pdfr2A-GAL4*(#84684), *fru-GAL4*(#66696), *dsx-GAL4*(#63434), *forMI01791-TG4.1* (#76140), *UAS-Pdfr-RNAi*(#42508), *UAS-myrGFP*(#77479), *UAS-for*(#37776), *for-RNAi*(#35158/#50741) were obtained from the Bloomington *Drosophila* Stock Center at Indiana University. *UAS(FRT.stop)mCD8GFP; fruFLP*(K1119), *LexAop-CD8GFP(II); UAS-CaLexA, LexAop-CD2-GFP* (K1234, originally BDSC #66542) were obtained from the Korea *Drosophila* Resource Center. To independently label cell membranes from CaLexA-driven GFP signals, we generated a transgenic strain (*UAS-mCD8RFP, lexAop-mCD8GFP* (X); *LexAop-CD8GFP* (II); *UAS-CaLexA, LexAop-CD2GFP* (III) by integrating BDSC stock #32229 and #66542. This design leverages distinct expression systems (UAS vs. LexAop) to separate RFP (membrane marker) and GFP (calcium activity reporter) signals. We thank Dr. Maria B. Sokolowski (University of Toronto) for sharing *forR* variants, Drs. Yuh Nung Jan and Lily Yeh Jan (UCSF, USA) for kindly sharing *GAL14-94* fly strain, Dr. Susan C.P. Renn (Washington University School of Medicine) for sharing *210y-GAL4* stain, Dr. Lihua Jin (Northeast Forestry University) for sharing *Hand-GAL4* stain.

## Mating duration assays

The mating duration assay in this study has been reported [15,16,19]. To enhance the efficiency of the mating duration assay, we utilized the *Df(1)Exel6234* (DF here after) genetic modified fly line in this study, which harbors a deletion of a specific genomic region that includes the sex peptide receptor (SPR) [88,89]. Previous studies have demonstrated that virgin females of this line exhibit increased receptivity to males [89]. We conducted a comparative analysis between the virgin females of this line and the CS virgin females and found that both groups induced SMD. Consequently, we have elected to employ virgin females from this modified line in all subsequent studies. For naïve males, 40 males from the same strain were placed into a vial with food for 5 days. For single reared males, males of the same strain were collected individually and placed into vials with food for 5 days. For experienced males, 40 males from the same strain were placed into a vial with food for 4 days then 80 DF virgin females were introduced into vials for last 1 day before assay. 40 DF virgin females were collected from bottles and placed into a vial for 5 days. These females provided both sexually experienced partners and mating partners for mating duration assays. At the fifth day after eclosion, males of the appropriate strain and DF virgin females were mildly anaesthetized by $CO_2$. After placing a single female into the mating chamber, we inserted a transparent film then placed a single male to the other side of the film in each chamber. After allowing for 1 h of recovery in the mating chamber in 25°C incubators, we removed the transparent film and recorded the mating activities. Only those males that succeeded to mate within 1 h were included for analyses. Initiation and completion of copulation were recorded with an accuracy of 10 sec, and total mating duration was calculated for each couple. All assays were performed from noon to 4pm. We conducted blinded studies for every test. The genetic controls using *GAL4*/ + or *UAS*/ +lines

were not included in the supplementary figures because we have previously demonstrated that 100% of these flies exhibit normal LMD and SMD behaviors [15,16,19]. Consequently, we only included additional genetic control experiments when they were considered essential for key fly lines.

## Immunostaining and antibodies

The dissection and immunostaining protocols for the experiments are described elsewhere [16,19]. After 5 days of eclosion, the *Drosophila* brain has been taken from adult flies and fixed in 4% formaldehyde at room temperature for 30 minutes. The sample has been washed three times (5 minutes each) in 1% PBT and then blocked in 5% normal goat serum for 30 minutes. The samples were first incubated overnight at 4°C with primary antibodies diluted in 1% PBT. Subsequently, fluorophore-conjugated secondary antibodies were applied and incubated for one hour at room temperature. The brain was mounted on plates with an antifade mounting solution (Solarbio) for imaging purposes. Samples were imaged with Zeiss LSM880. Primary antibodies: Chicken anti-GFP (1:500, Invitrogen), rabbit anti-LacZ antibody (1:1000, Rockland), rat anti-elav (1:100, DSHB), mouse anti-Bruchpilot (nc82) (1:50, DSHB). Fluorophore-conjugated secondary antibodies: Alexa-488 donkey anti-chicken (1:200, Jackson), Alexa Fluor 488-conjugated goat anti-rabbit (1:100, Thermo), Alexa Fluor 555-conjugated donkey anti-rat (1:100, Thermo), Alexa Fluor 647-conjugated goat anti-mouse (1:100, Jackson), plasma membranes of fatbody (stained by CellMask Deep red C10046, Thermo).

## Quantitative analysis of fluorescence intensity

To ascertain calcium levels from microscopic images, we dissected and imaged five-day-old flies of various social conditions and genotypes under uniform conditions. For group reared (naïve) condition, the flies were reared in the group condition and dissect right after 5 days of rearing without any further action. For single reared condition, the flies were reared in single condition and dissect at the same time as group reared flies right after 5 days of rearing. The GFP signal in the brains and hearts was amplified through immunostaining with chicken anti-GFP primary antibody. Image analysis was conducted using ImageJ software. For the quantification of fluorescence intensities, an investigator, blinded to the fly's genotype, thresholded the sum of all pixel intensities within a sub-stack to optimize the signal-to-noise ratio, following established method [90]. The total fluorescent area or region of interest (ROI) was then quantified using ImageJ, as previously reported. For CaLexA signal quantification, we adhered to protocols detailed by Kayser et al. [91], which involve measuring the ROI's GFP-labeled area by summing pixel values across the image stack. This method assumes that changes in the GFP-labeled area are indicative of alterations in the CaLexA signal, reflecting synaptic activity. CaLexA signals were quantified using 'integrated signal density' (summed pixel intensity within manually defined ROIs), not merely ROI area. ROI intensities were background-corrected by measuring and subtracting the fluorescent intensity from RFP channel or nc82 channel, as per Kayser et al. [91].

## Single-nucleus RNA-sequencing analyses

snRNAseq dataset analyzed in this paper is published in [47] and available at the Nextflow pipelines (VSN, https://github.com/vib-singlecell-nf), the availability of raw and processed datasets for users to explore, and the development of a crowd-annotation platform with voting, comments, and references through SCope (https://flycellatlas.org/ scope), linked to an online analysis platform in ASAP (https://asap.epfl.ch/fca).

## Statistical analysis

Statistical analysis of mating duration assay was described previously [19]. More than 50 males (naïve, experienced and single) were used for mating duration assay. Our experience suggests that the relative mating duration differences between naïve and experienced condition and singly reared are always consistent; however, both absolute

values and the magnitude of the difference in each strain can vary. So, we always include internal controls for each treatment as suggested by previous studies [92]. Therefore, statistical comparisons were made between groups that were naïvely reared, sexually experienced and singly reared within each experiment. As mating duration of males showed normal distribution (Kolmogorov-Smirnov tests, $p > 0.05$), we used two-sided Student's t tests. The mean ± standard error (s.e.m) (**** = $p < 0.0001$, *** = $p < 0.001$, ** = $p < 0.01$, * = $p < 0.05$). All analysis was done in GraphPad (Prism). Individual tests and significance are detailed in figure legends. Besides traditional t-test for statistical analysis, we added estimation statistics for all MD assays and two group comparing graphs. In short, 'estimation statistics' is a simple framework that—while avoiding the pitfalls of significance testing—uses familiar statistical concepts: means, mean differences, and error bars. More importantly, it focuses on the effect size of one's experiment/intervention, as opposed to significance testing [93]. In comparison to typical NHST plots, estimation graphics have the following five significant advantages such as (1) avoid false dichotomy, (2) display all observed values (3) visualize estimate precision (4) show mean difference distribution. And most importantly (5) by focusing attention on an effect size, the difference diagram encourages quantitative reasoning about the system under study [94]. Thus, we conducted a reanalysis of all of our two group data sets using both standard t tests and estimate statistics. In 2019, the Society for Neuroscience journal eNeuro instituted a policy recommending the use of estimation graphics as the preferred method for data presentation [95].

## Supporting information

**S1 Fig. Critical Role of *foraging* Alleles in LMD and SMD behaviors.** (A-B) LMD and SMD assays for *for^R*/+ and *for^S*/+ males. (C) LMD assay of flies expressing *elav^c155* drives together with *for-RNAi^GL00026*. (D) LMD assay for *for-RNAi^HSM04486*/+. (E) LMD assay of flies for *elav^c155*/Y. (F) LMD assay of flies for *Pdfr^2A-GAL4*/+. (G) LMD assay of flies expressing *nSyb-GAL4* together with *for-RNAi.*
(TIF)

**S2 Fig. LMD Requires Specific EB Neurons with *foraging* Gene.** (A-K) LMD assays for male overexpressing *foraging* driven by subsets of neuronal cell drivers of the EB, *30y-GAL4*(A), *078y-GAL4* (B), *210y-GAL4* (C), *104y-GAL4* (D), *007y-GAL4* (E), *c42-GAL4* (F), *c232-GAL4* (G), *c205-GAL4* (H), *c819-GAL4* (I), *c161-GAL4* (J), and *c61-GAL4* (K). (L) LMD assay for *30y-GAL4* drives *for* overexpression under *for^s* homozygote. (M) LMD assay for *UAS-for*/+.
(TIF)

**S3 Fig. The co-expression of *foraging* gene and Pdfr in various tissues.** (*A*-N) SCOPE scRNA-seq datasets reveal cell clusters colored by expression of *foraging* (blue), *Pdfr* (green) in different tissues. (O) Male flies expressing the *for^TG4.1*-GAL4 together with *UAS-mCD8GFP* were immunostained with anti-GFP (green) antibody and phalloidin (magenta). (P) Male flies brain expressing the *UAS(FRT.stop)mCD8GFP; fru^FLP* together with *Pdfr^2A-GAL4* were immunostained with anti-GFP (green) and nc82 (magenta) antibodies. Scale bars represent 100 µm. (Q) Male flies heart expressing the *UAS(-FRT.stop)mCD8GFP; fru^FLP* together with *30y-GAL4* were immunostained with anti-GFP (green) antibody and phalloidin (magenta). Scale bars represent 100 µm.
(TIF)

**S4 Fig. *foraging* neurons in the AL of the brain are responsible for regulating LMD.** (A) Different levels of neural activity of the brain as revealed by the CaLexA system in group and single reared flies. Male flies expressing *for^MI01791-TG4.1* along with *LexAop-CD8GFP(II); UAS-CaLexA, LexAop-CD2-GFP* were dissected after 5 days of growth. The dissected brains were then immunostained with anti-GFP (green) and anti-nc82 (blue). GFP is pseudo-colored as "red hot". Scale bars represent 100 µm in brain panels. (B) The GFP fluorescence (green) in male fly brain was processed using ImageJ software, where a threshold function was applied to distinguish fluorescence from the background. (C) Quantification of

relative value for GFP fluorescence. (D) Different levels of intracellular calcium level of the fat body as revealed by the CaLexA system in group and single reared flies. Scale bars represent 25 μm in fat body panels. (E) Quantification of mean intensity for GFP fluorescence.
(TIF)

**S5 Fig.** *foraging* **Gene's function Independent of Serotonergic system in LMD behavior.** (A-B) LMD assays of flies expressing *5-HT1B-GAL4* drives together with *for-RNAi* (A) *and UAS-for* (B). (C-D) LMD assays of flies expressing *Trhn-GAL4* drives together with *for-RNAi* (C) *and UAS-for* (D). (E) SCOPE scRNA-seq datasets reveal cell clusters colored by expression of *foraging* (blue), *5-HT1B* (green) in neurons. (F) SCOPE scRNA-seq datasets reveal cell clusters colored by expression of *foraging* (blue), *Trhn* (green) in neurons.
(TIF)

**S1 File.  Calcium signals (GFP) in** *Drosophila* **heart cells (Hand-GAL4, RFP). Phalloidin stained in purple.**
(JPG)

## Acknowledgments

We are very appreciative to the colleagues who supplied us with several fly strains. We thank Dr. Maria B. Sokolowski (University of Toronto) for sharing *for^R* variants, Drs. Yuh Nung Jan and Lily Yeh Jan (UCSF, USA) for kindly sharing *GAL^{14-94}* fly strain, Dr. Susan C.P. Renn (Washington University School of Medicine) for sharing *210y-GAL4* stain, Dr. Lihua Jin (Northeast Forestry University) for sharing *Hand-GAL4* stain.

## Author contributions

**Formal analysis:** Woo Jae Kim.

**Funding acquisition:** Woo Jae Kim.

**Investigation:** Hongyu Miao, Wenjing Li, Yongwen Huang, Woo Jae Kim.

**Methodology:** Woo Jae Kim.

**Project administration:** Woo Jae Kim.

**Resources:** Woo Jae Kim.

**Visualization:** Woo Jae Kim.

**Writing – original draft:** Woo Jae Kim.

**Writing – review & editing:** Hongyu Miao, Wenjing Li, Yongwen Huang, Woo Jae Kim.

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
