## [Decision Letter · Decision Letter 0]

PGENETICS-D-25-00332

Heart-Specific  foraging  Gene Function Bridges Social Cues and Interval Timing Behaviors in  Drosophila

PLOS Genetics

Dear Dr. Kim,

Thank you for submitting your manuscript to PLOS Genetics. After careful consideration, we feel that it has merit but needs some additional, minor, revisions to meet PLOS Genetics's publication criteria. We invite you to submit a revised version of the manuscript that addresses the points raised during the review process.

Please submit your revised manuscript within 30 days May 25 2025 11:59PM. If you will need more time than this to complete your revisions, please reply to this message or contact the journal office at plosgenetics@plos.org. Please include the following items when submitting your revised manuscript:

We look forward to receiving your revised manuscript.

Kind regards,

Mariana Federica Wolfner

Academic Editor

PLOS Genetics

Monica Colaiácovo

Section Editor

PLOS Genetics

Aimée Dudley

Editor-in-Chief

PLOS Genetics

Anne Goriely

Editor-in-Chief

PLOS Genetics

**Additional Editor Comments:**

Thank you for this very nice revised manuscript. It has been reviewed by two of the previous experts, and one additional expert since the third original reviewer was unavailable. All three experts are very positive about the paper. They recommend some minor revisions that you are encouraged to consider and make. The revised manuscript will not be sent for re-review before the decision is recorded.

**Journal Requirements:**

1) Please provide an Author Summary. This should appear in your manuscript between the Abstract (if applicable) and the Introduction, and should be 150-200 words long. The aim should be to make your findings accessible to a wide audience that includes both scientists and non-scientists. Sample summaries can be found on our website under Submission Guidelines:

https://journals.plos.org/plosgenetics/s/submission-guidelines#loc-parts-of-a-submission

3) Please amend your detailed Financial Disclosure statement. This is published with the article. It must therefore be completed in full sentences and contain the exact wording you wish to be published. Please ensure that the funders and grant numbers match between the Financial Disclosure field and the Funding Information tab in your submission form. Note that the funders must be provided in the same order in both places as well.

**Reviewers' comments:**

Reviewer's Responses to Questions

**Comments to the Authors:**

Reviewer #1: The authors have substantially revised the manuscript. The revised manuscript elucidates an important role of the Drosophila foraging (for) gene in bridging social cues and interval timing behaviors through dual neural and non-neural (cardiac) mechanisms. Combining behavioral assays, single-cell RNA sequencing, gene knockdown, and calcium imaging, the study provides novel insights into the genetic and environmental interplay underlying adaptive behaviors. While the manuscript has been significantly improved, I still have some minor concerns on the writing and intepretation of data.

Minor concerns:

1. language issues. Remove claims such as 'novel'. Longer mating duration (LMD) has been repeatly dipicted such as in line 198, and wrongly stated in line 289.

2. wrong lable of color in Figure 6E and 6G.

3. Regarding to the regular phenotype in forS/+ and forR/+ flies, but deficit LMD and SMD in forR/forS flies, I think an alternative explanation for this discrepency is that there are potential mutations other than the for gene in forR and forS flies, which are dominant in forR/forS flies, but recessive in forS/+ and forR/+ flies.

Reviewer #2: I thank the authors for the improvements in the manuscript. I am happy that within a short time they found Pdfr+ for+ (possibly fru+ too) cells, somewhat unexpectedly, in the adult heart, playing an important role in LMD. Given their complete absence from the earlier version (V1) of the manuscript, I urge the authors to ensure that these heart-related new findings are derived from meticulous and rigorous experiments with sufficient sample sizes and repeats, and conservative interpretations. There are still a few minor errors (like Pdfr3A-gal4 in Fig 3 should be Pdfr-2A-Gal4). Lastly, I fail to understand in Fig 6. where does the RFP signal come from - it's unclear from the written genotypes. Please clarify if you measured integrated signal density or just the area of the manually chosen ROI to quantify CaLexA - it's not clear from the "Mat & Met".

Reviewer #3: The review is uploaded as an attachment where the italics are maintained but I also paste it here for convenience.

Manuscript number: RC-2024-02605

Corresponding author: Woo Jae, Kim

I was excited to read this paper as it addresses and brings together aspects of research around the role of the foraging gene that has remained refractory. The authors of the paper accomplish this with state-of-the-art techniques (e.g. split gal4, brain scRNAseq, outstanding design and control of Drosophila behavior-genetic experiments). The phenotype under study, the interval timing of mating duration, is fascinating as it is an adaptive behaviour responsive to environmental input (e.g. previous social and mating experience, metabolic state, current social environment during the mating duration test). How our experience gets embedded into our biology from a gene regulatory and circuit perspective has broad significance to a variety of disciplines. Finally, because the foraging gene orthologues have been shown to play a role in behavioural tasks in eusocial insects, other invertebrates and vertebrates, the present study will be of interest to an audience that extends outside of Drosophila.

A role for the foraging gene in any aspect of D. melanogaster mating behavior has not been investigated. Furthermore, little has been done to map the circuits that foraging is involved with for any of its pleiotropic traits, except for the latency of the larval nociception response (see Dason et al 2020, PNAS). Additionally, as mentioned by the authors, although foraging’s effects on GEI/plasticity in metabolic and behavioural responses of the rover/sitter variants have been shown, an understanding of how these effects might be regulated through tissue-specific functions is lacking. The authors of this manuscript are the first to identify foraging’s critical role in interval timing of mating duration, an important adaptive phenotype. Their results are convincing. They have identified two organs, subsets of cells in the heart and the ellipsoid body in the brain that together link foraging’s metabolic and neuronal/glial effects on mating duration. Furthermore, they have shown that foraging interacts with the “timing” gene pdfr in the brain and with the sex determination gene doublesex, in the heart to affect interval timing in mating. foraging’s role in plastic responses to the social environment in the heart is found when comparing male flies with and without social experience. Future research on this model might interrogate how environmental information to the heart gets transduced to the brain to affect interval timing on mating duration under different social and metabolic conditions. Future research could also address how foraging’s molecular complexity affects interval timing in mating behavior; foraging is a highly complex gene with 4 promoters, 21 transcripts and 8 protein isoforms. Mapping the modularity of the phenotypes under study, the effects of social environments, their tissue specificity and sex differences to foraging’s promotors, transcripts, proteins and gene regulatory networks, along with further circuit analysis will continue to develop the model studied in the current paper. I look forward to reading future papers from this group on this model.

Below are some further comments on your paper. Most of them are minor.

Introduction:

I like the introduction. It is thorough and well-written.

I want to point out that the title of the paper is specific, only representing the heart-specific function. Do the authors want the title to more broadly reflect the findings in the paper?

The paper Reaume CJ, Sokolowski MB, Mery F. 2011. A natural genetic polymorphism affects retroactive interference in Drosophila melanogaster. 2011 Proc Biol Sci. 278(1702):91-8. doi: 10.1098/rspb.2010.1337, may be of interest. It addresses retroactive interference/forgetting, linking, foraging behaviour, timing and memory in the rover/sitter variants.

Results:

Line 143, you state that “Rovers, which have one copy of the foraging gene (forR/-), exhibit higher levels of for-mRNA and PKG activity compared to sitters, who are homozygous for the loss-of-function allele (forS/forS)”. This is not correct. Rovers have 2 copies of the forR allelic variants-it is homozygous for forR. Furthermore, sitters are not loss of function alleles. They are hypomorphs and the sitter has two copies of the forS allele. Loss of function alleles of foraging are pupal lethal. See Allen et al 2017 Genetics. Please correct this. I don’t think these errors affect the results or interpretations in your paper.

Line 150-152, reword “neurons of sitter allele”. As you know alleles don’t have neurons. Say neurons carrying (or with) the sitter alleles or something like that. Also, correct for when you say “neurons of rover allele” in the next sentence.

Line 154 “the axon terminal projections of sitter strains are altered”. Expand on this. Say how they are altered in their branching patterns at the larval neuromuscular junction (nmj). See also Dason et al 2019 J. Cell Science, and Dason and Sokolowski 2021 J. Neurogenet for further analyses that agree and expand on the 1999 Renger et al paper and provide information on foraging's pleiotropic effects at the synapse.

Line 161 Regarding the results of the forR/fors heterozygotes, I think that this is a very interesting result, a kind of co-dominance in the heterozygote. For a future study, it would be interesting to see if LMD and SMD behaviours are differentially regulated through each of foraging's 4 promoters.

Line 167-169, “Therefore, we hypothesize that an extremely high level of PKG activity specifically disrupts SMD, while an extremely low level of PKG activity specifically disrupts LMD behavior.” I think that this is an interesting idea. Are there other examples of this in the literature? I recollect that the fly dunce gene might work like this in learning and memory but can’t recall the paper.

Line 196, see also Kuntz S, Poeck B, Sokolowski MB, Strauss R. 2012 The visual orientation memory of Drosophila requires Foraging (PKG) upstream of Ignorant (RSK2) in ring neurons of the central complex. Learn Mem. 19(8):337-40. doi: 10.1101/lm.026369.

Lines 233 and 323, These are exciting results!

Line 393, “with the rover variant specifically affecting SMD and the sitter variant affecting SMD.” I think there is a typo one should say LMD.

Lines 443-445. There are many innovative aspects of this paper and lots of new knowledge that was not previously attainable because of the lack of tools available.

Line 513 reword, “The sample has been next be incubated” ?

Line 527 reword, the flies were reared in the group condition and dissected right after 5 days of rearing.

Line 529. the phrase “without after any action” is not necessary in this sentence or the one below this sentence.

Line 555-557, “Therefore, statistical comparisons were made between groups that were naïvely reared, sexually experienced and singly reared within each experiment.” The point “reared within each experiment” in this sentence makes perfect sense to me.

Figures: The SCope figures in the paper are difficult to read because there are no keys to tell us what the overlapping colours would be. Often in your figures, three genes are compared in for example, red, green and blue and when looking at the image, it is difficult to see which gene coexpresses with which other gene. This is true for most of your Scope representations. If you provided a key for colour of the overlaps (not just the colour for each gene on the axes), the results would be clear to the reader. The keys could be put in the figure captions or next to the figure. My experience is that when we make readers do too much work while looking at figures, many will give up. Another suggestion about the colours in your SCope figures is that they don’t work for people who are colour-blind. Whether this requires changes in your paper is, of course, up to the editor. Your images with gfp (green and purple) are fine in this regard.

**Have all data underlying the figures and results presented in the manuscript been provided?**

Reviewer #1: Yes

Reviewer #2: Yes

Reviewer #3: Yes

PLOS authors have the option to publish the peer review history of their article (what does this mean? ). If published, this will include your full peer review and any attached files.

**Do you want your identity to be public for this peer review?** For information about this choice, including consent withdrawal, please see our Privacy Policy .

Reviewer #1: No

Reviewer #2: **Yes: ** Abhishek Chatterjee

Reviewer #3: No

**Figure resubmission:**
---

## [Editor Report · Decision Letter 1]

Dear Dr Kim,

We are pleased to inform you that your manuscript entitled "The foraging  Gene Coordinates Brain and Heart Networks to Modulate Socially Cued Interval Timing in Drosophila" has been editorially accepted for publication in PLOS Genetics. Congratulations!

Yours sincerely,

Mariana Federica Wolfner

Academic Editor

PLOS Genetics

Monica Colaiácovo

Section Editor

PLOS Genetics

Aimée Dudley

Editor-in-Chief

PLOS Genetics

Anne Goriely

Editor-in-Chief

PLOS Genetics

Comments from the reviewers (if applicable):

Thank you for the thorough revisions, and for the chance to consider this exciting paper.

**Data Deposition**

http://datadryad.org/submit?journalID=pgenetics&manu=PGENETICS-D-25-00332R1

**Press Queries**

---

## [Editor Report · Acceptance letter]

PGENETICS-D-25-00332R1

The foraging Gene Coordinates Brain and Heart Networks to Modulate Socially Cued Interval Timing in Drosophila

Dear Dr Kim,

We are pleased to inform you that your manuscript entitled "The foraging Gene Coordinates Brain and Heart Networks to Modulate Socially Cued Interval Timing in Drosophila " has been formally accepted for publication in PLOS Genetics! Your manuscript is now with our production department and you will be notified of the publication date in due course.

With kind regards,

Anita Estes

PLOS Genetics

On behalf of:
